# Two-dimensional semiconducting SnP$_2$Se$_6$ with giant second-harmonic-generation for monolithic on-chip electronic-photonic integration

Cheng-Yi Zhu[1,6], Zimeng Zhang[2,6], Jing-Kai Qin [1] ✉, Zi Wang[2], Cong Wang[3], Peng Miao[4], Yingjie Liu[2], Pei-Yu Huang[1], Yao Zhang[2], Ke Xu [2] ✉, Liang Zhen [1,5], Yang Chai [3] ✉ & Cheng-Yan Xu [1,5] ✉

Two-dimensional (2D) layered semiconductors with nonlinear optical (NLO) properties hold great promise to address the growing demand of multi-function integration in electronic-photonic integrated circuits (EPICs). However, electronic-photonic co-design with 2D NLO semiconductors for on-chip telecommunication is limited by their essential shortcomings in terms of unsatisfactory optoelectronic properties, odd-even layer-dependent NLO activity and low NLO susceptibility in telecom band. Here we report the synthesis of 2D SnP$_2$Se$_6$, a van der Waals NLO semiconductor exhibiting strong odd-even layer-independent second harmonic generation (SHG) activity at 1550 nm and pronounced photosensitivity under visible light. The combination of 2D SnP$_2$Se$_6$ with a SiN photonic platform enables the chip-level multi-function integration for EPICs. The hybrid device not only features efficient on-chip SHG process for optical modulation, but also allows the telecom-band photodetection relying on the upconversion of wavelength from 1560 to 780 nm. Our finding offers alternative opportunities for the collaborative design of EPICs.

Complementary metal–oxide–semiconductor (CMOS) compatible silicon photonic chips promises to address the growing demand of multifunction integration for next-generation optoelectronic circuitries and systems. Owing to the excellent bandgap tunability, strong light-matter interaction and high materials compatibility, atomically thin two-dimensional (2D) layered semiconductors and related hybrid structures have demonstrated plenty of fascinating phenomena and ground-breaking technological applications in optoelectronics[1,2]. The possibility of constructing van der Waals (vdW) heterostructures

without considering conventional 'lattice mismatch' issue allows 2D semiconductors to be easily integrated into silicon photonics platform for high-performance photodetector operated in telecom band (1310 – 1550 nm)[3]. For the 2D semiconductors with the capability of nonlinear optical (NLO) wavelength conversion, and in particular, the second-harmonic generation (SHG), output signal can be generated with frequency doubled from the incident photon field, which occurs in the crystal with intrinsically broken inversion symmetry[4–6]. The hybrid silicon photonics integrated with 2D NLO materials have proven

[1]Sauvage Laboratory for Smart Materials, School of Materials Science and Engineering, Harbin Institute of Technology (Shenzhen), Shenzhen 518055, China. [2]Guangdong Provincial Key Laboratory of Semiconductor Optoelectronic Materials and Intelligent Photonic Systems, Harbin Institute of Technology (Shenzhen), Shenzhen 518055, China. [3]Department of Applied Physics, The Hong Kong Polytechnic University, Hong Kong, China. [4]HORIBA Scientific, Shanghai 205335, China. [5]MOE Key Laboratory of Micro-Systems and Micro-Structures Manufacturing, Harbin Institute of Technology, Harbin 150080, China. [6]These authors contributed equally: Cheng-Yi Zhu, Zimeng Zhang. ✉e-mail: jk.qin@hit.edu.cn; kxu@hit.edu.cn; ychai@polyu.edu.hk; cy_xu@hit.edu.cn

to be an ideal platform for the implementation of on-chip optical modulation and signal processing[7,8]. By combining the strong NLO activity with excellent optoelectronic properties of 2D NLO semiconductors, one can expect the accomplishment of integrated multifunctions required for monolithic on-chip electronic-photonic integrated circuits (EPICs), such as light generation[9,10], frequency conversion[11,12], nonlinear electro-optic modulation[13,14], photodetection and compact on-chip optoelectronic integration[3,15,16].

A series of 2D layered semiconducting materials with excellent NLO characteristics in the visible–infrared (IR) spectral range have been reported[17,18]. Monolayer transition metal dichalcogenides (TMDCs), such as $MoS_2$, $WS_2$, $MoSe_2$ and $WSe_2$, possess high second-order nonlinear coefficients due to their broken inversion symmetry[5,19]. The semiconducting TMDCs in 2H phase can be obtained with large enough size for EPICs applications. However, the SH polarization is highly confined in odd layers since the inversion symmetry can be restored in even-layered samples, leading to an oscillatory and degenerated SH response with the increasing of layer numbers[20,21]. To guarantee enough light-matter interaction length and efficient light-absorption, 2H-stacked TMDCs integrated into silicon photonics shall be of considerable thickness. The dilemma between restoration of inversion symmetry and sufficiently long light-matter interaction drastically dilutes the applicability of 2H-stacked TMDCs. The inversion asymmetry can be well maintained with layer increasing in few-layer TMDCs stacked in 3 R configuration, which contributes to giant SHG activity for on-chip nonlinear optical devices[22–24]. However, we noted that previously-reported 2D NLO semiconductors usually demonstrate barely satisfactory photoelectric response and relatively low SHG susceptibility in standardized telecom bands ($<1 \times 10^{-10}$ m·V$^{-1}$), far from satisfying the requirements of chip-level electronic-photonic co-design for on-chip telecommunication[5,19].

In this work, we report a 2D layered $SnP_2Se_6$ semiconductor with high SHG susceptibility ($-1.3 \times 10^{-9}$ m·V$^{-1}$) at 1550 nm wavelength. Due to the disordered interlayer stacking mode, the broken inversion symmetry of $SnP_2Se_6$ crystal can retain in multilayer samples, which contributes to odd-even layer-independent SHG response and superposition of SHG signals in thick samples. In addition, 2D $SnP_2Se_6$ exhibits excellent electrical and optoelectronic properties in ambient condition, and the photodetector features high photoresponsivity of $10^3$ A W$^{-1}$ and fast response rate of 412 µs under visible light. Furthermore, we experimentally demonstrated a prototype on-chip hybrid device by combining 2D $SnP_2Se_6$ into SiN photonic platform. The giant SHG activity and promising optoelectronic characteristics enable the device to operate with the monolithically integrated functions of converting and detecting optical signals at 1560 nm. These findings demonstrate the fascinating physical properties of emerging 2D $SnP_2Se_6$ NLO semiconductor, making it a promising candidate for applications in on-chip telecom-wavelength conversion and photodetection.

## Results

### Synthesis and microstructure characterization

High-quality ultrathin $SnP_2Se_6$ nanosheets were synthesized using the space-confined chemical vapor transport (SCCVT) method (See Methods and Supplementary Fig. 1 for details). $SnP_2Se_6$ belongs to space group $R3$, and it has a typical vdW layered structure with an interlayer spacing of 0.68 nm. The basic unit of single-layer $SnP_2Se_6$ is composed of non-stereoactive $Sn^{4+}$ cations, $(P_2Se_6)^{4-}$ anions and vacancies, showing a crystal structure with high non-centrosymmetry (Fig. 1a, b). The as-grown $SnP_2Se_6$ nanosheets on mica substrate reveal typical hexagonal shapes, and the thickness can be scaled down to 0.7 nm. More importantly, large-area single-crystalline $SnP_2Se_6$ nanosheets can also be achieved, with the maximum lateral size reaching up to millimeters (Fig. 1c and Supplementary Fig. 2). In the high-resolution X-ray photoelectron spectroscopy (XPS) spectra

(Supplementary Fig. 3), Sn $3d$, P $2p$ and Se $3d$ orbitals are clearly revealed, with a stoichiometric ratio of 1:2:5.95 by fitting the integrated peak areas. The results demonstrate the synthesis and ideal atomic stoichiometry of the $SnP_2Se_6$ nanosheets.

Figure 1d depicts the Raman spectra of samples with different thickness. 2D $SnP_2Se_6$ reveals seven obvious Raman peaks: P1 (~82 cm$^{-1}$), P2 (~119 cm$^{-1}$), P3 (~156 cm$^{-1}$), P4 (~185 cm$^{-1}$), P5 (~218 cm$^{-1}$), P6 (~433 cm$^{-1}$), and P7 (~447 cm$^{-1}$). P1 to P4 located below 200 cm$^{-1}$ represent the external and internal Se-P-Se bending mode of the $PSe_3$ structural units, while P5, P6 and P7 are associated with the $Se_3P$-$PSe_3$ and P-Se valence vibrations[25]. We do not observe obvious frequency shift of Raman modes with the increasing of layer numbers, which might be attributed to the weak interaction between layers. The angular dependence of the Raman intensity exhibits a characteristic 6-fold rotational symmetry, indicating the 2D $SnP_2Se_6$ are isotropic in the layer plane (Supplementary Fig. 4). Raman intensity mappings of P3 and P5 peaks were also performed on a trigonal sample with thickness about 12 nm (Inset images in Fig. 1d). The uniform distribution demonstrates the high crystalline quality of $SnP_2Se_6$ nanosheets.

The optical absorption properties of 2D $SnP_2Se_6$ was investigated using the micro-UV-visible absorption spectroscopy (Fig. 1e). The value of band gap for 2 L $SnP_2Se_6$ nanosheet is estimated to be 1.93 eV from the Tauc curve (inset image in Fig. 1e), and it would drop to 1.76 eV as the layer increases to 16 L. Although density functional theory (DFT) calculations usually underestimate the band gaps, the trend of change agrees well the experimental results. As plotted in Fig. 1f and Supplementary Fig. 6, monolayer $SnP_2Se_6$ has an indirect bandgap of 1.7 eV, with the conduction band minimum (CBM) and valence band maximum (VBM) both locating at the $\Gamma$ and $K$ points of the 2D Brillouin zone, respectively. The indirect bandgap would drop to 1.58 eV for the bilayer $SnP_2Se_6$ and further be decreased down to 1.52 eV for bulk counterpart.

With the assistance of aberration-corrected high-angle annular dark-field scanning transmission electron microscopy (AC HAADF-STEM), the atomic structure and lattice information of 2D $SnP_2Se_6$ was successfully revealed. The energy dispersive spectroscopy (EDS) elemental mapping images demonstrate the homogeneous distribution of Sn, P and Se elements (Supplementary Fig. 5). The plan-view image identifies a highly-crystallized structure without any atomic vacancies (Fig. 1g). By combining a closer examination of the atomic arrangement pattern with corresponding selected-area electron diffraction (SAED) patterns, we can identify the equal interplanar spacing of 3.30 Å for (110) and ($\bar{1}$20) planes, which have a crossing angle of 60°. The HAADF intensity profile, which is directly related to the averaged atomic number, was also taken along the purple dashed line I and purple dashed line II, respectively. The periodic variation in the relative intensities indicates the patterned arrangement of Sn, P, and Se atoms, and the bonding length of the Sn-Sn atoms can be determined to be 6.50 Å. As illustrated in Fig. 1h, the distance of nearby Se-P pairs changes from 2.2 to 2.4 Å, which means the orderly distributed P-P pairs shift slightly off the symmetric center of the selenium skeleton. It will lead to the formation of structurally distorted $[SnSe_6]^{8-}$ and $[P_2Se_6]^{4-}$ octahedrons and highly distorted rhombohedral structure of $SnP_2Se_6$ crystals.

### Giant SHG response at telecom wavelength

NLO crystals operated in telecom band are of crucial applications for on-chip frequency conversion and signal processing. As a result of the broken in-plane inversion symmetry, 2D $SnP_2Se_6$ crystals are expected to deliver large SHG response. As depicted in Fig. 2a, SHG signal is collected from individual $SnP_2Se_6$ nanosheet in the back-reflection configuration. Figure 2b presents the power-dependent SHG spectrum of an 8-nm-thick $SnP_2Se_6$ nanosheet under an excitation wavelength of 1550 nm. A peak at 775 nm can be clearly detected, which is exactly half

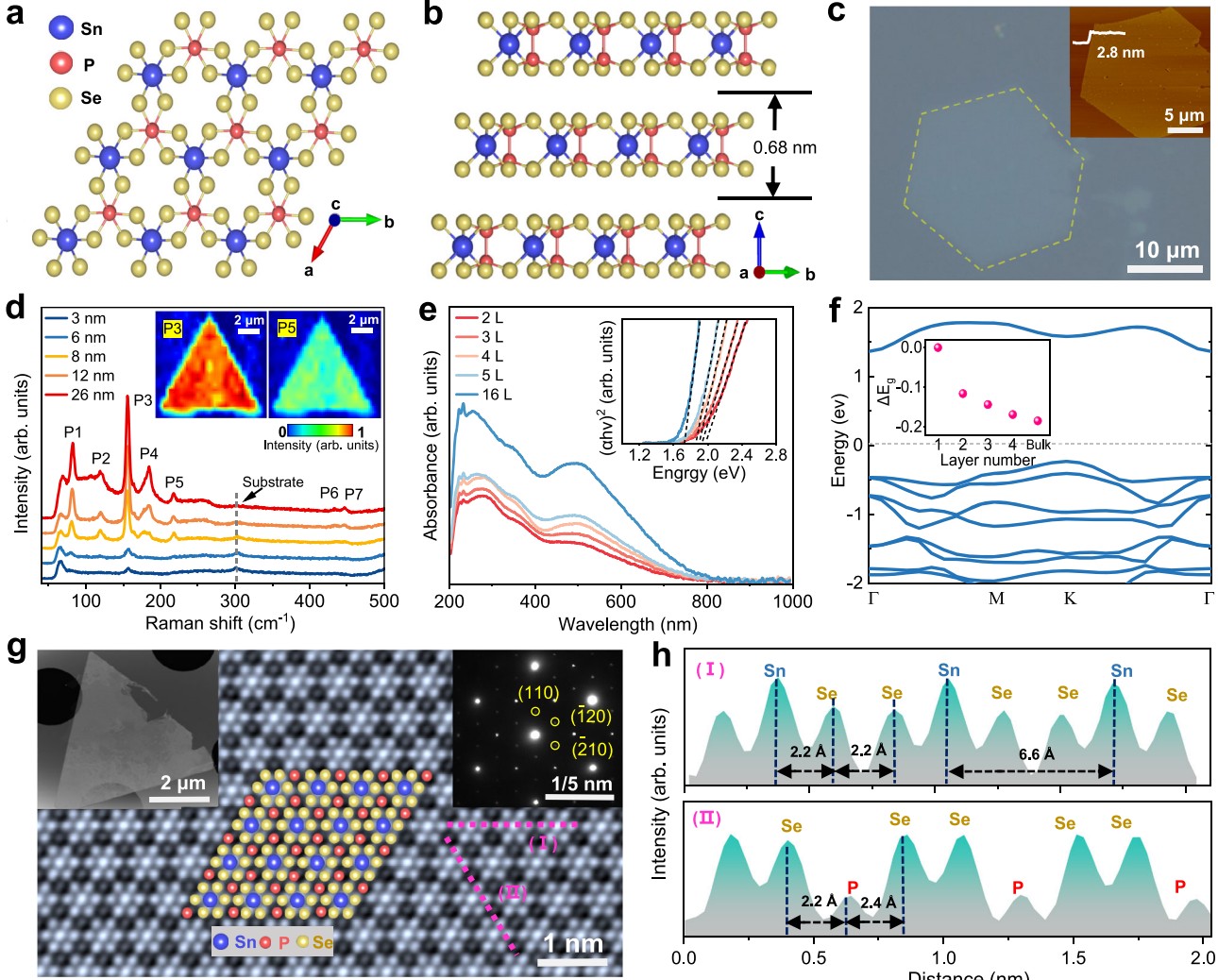

**Fig. 1 | Atomic structure of SnP₂Se₆ and material characterization. a, b** Atomic structure of 2D SnP₂Se₆. **c** Optical microscopy image of as-grown SnP₂Se₆ nanosheets on mica substrate. The hexagonal shape of the nanosheet is outlined by yellow dashed lines. Inset images shows the corresponding atomic force microscope (AFM) topography image and height profile of as-prepared SnP₂Se₆ nanosheets. **d** Thickness-dependent Raman spectra of SnP₂Se₆ nanosheets. Inset images exhibit Raman mapping of $P_3$ and $P_5$ characteristic peaks. **e** Thickness-dependent micro-UV-visible absorption spectra of SnP₂Se₆ nanosheets. Inset image exhibits optical band gap identified by the micro-UV-visible absorption spectroscopy. The $\alpha$, $h$, and $\nu$ in the ordinate of the Tauc plot represent the absorption coefficient of the material, Planck's constant, and the frequency of light,

respectively. **f** Calculated band structure of SnP₂Se₆ monolayer using the HSE06 functional. The horizontal dotted line indicates the Fermi level ($E_F = 0$ eV). The inset image displays the band gap variation ($\Delta E_g$) of SnP₂Se₆ nanosheets as the number of layers increases. **g** Aberration-corrected high-angle annular dark-field scanning transmission electron microscopy (HAADF-STEM) image of SnP₂Se₆ nanosheet, together with the corresponding top-view atomic model. The inset images present the low-magnification bright-field TEM image of triangular SnP₂Se₆ nanosheet (top left) and selected-area electron diffraction (SAED) pattern (top right). **h** Atomic intensity profiles of aberration-corrected HAADF-STEM image along the purple dashed lines in Fig. 1g.

of the incident wavelength. The SHG intensity scales quadratically with the pump intensity, and it can be well fitted with a high coefficient of 1.95 (Fig. 2c). Polarization-resolved SHG measurement providing the crystallographic information of the crystals was also performed (Fig. 2d and Supplementary Fig. 7). We find a characteristic 6-fold pattern, which exactly reflects the underlying three-fold rotational crystal symmetry of SnP₂Se₆ crystals. The result shows that one can directly identify the crystallographic direction of SnP₂Se₆ layers merely by SHG characterization. Supplementary Fig. 8 shows the SH mapping image of the same sample. The high uniformity of intensity distribution indicates that the layered SnP₂Se₆ nanosheet is single-crystalline in nature.

The application of 2D NLO materials for optical modulation requires the samples with considerable thickness to guarantee enough matter-light interaction length and light-absorption. The SHG intensity of 2D SnP₂Se₆ rises gradually with the increasing of layer number

(Fig. 3a), showing totally different characteristics from other 2D semiconducting 2H-TMDCs[26]. Here, 2H-MoTe₂ possessing among the highest second-order nonlinear susceptibility ($\chi^{(2)}$) in 2D layered semiconducting materials under 1550 nm, is utilized as the counterpart for comparison[27]. As shown in Supplementary Fig. 9, 2H-MoTe₂ also exhibits polarization-dependent SHG response like SnP₂Se₆. Although 2H-MoTe₂ exhibits a higher SHG intensity than SnP₂Se₆ in monolayer limit, its intensity would drop and fluctuate with the increasing of layer number, together with the odd-even dependencies (Fig. 3b). In contrast, SnP₂Se₆ nanosheets with few layers (<7 layers) demonstrates quadratically increased SH response, and the collected SHG peak intensity for multilayer SnP₂Se₆ nanosheets exceeds that of 2H-MoTe₂ with layer number increasing. The value of SHG intensity for 9-layer SnP₂Se₆ is almost an order of magnitude larger than that of 2H-MoTe₂ with the same thickness (Inset image in Fig. 3b). It should be noted that the influence of light re-absorption is being strengthened gradually in

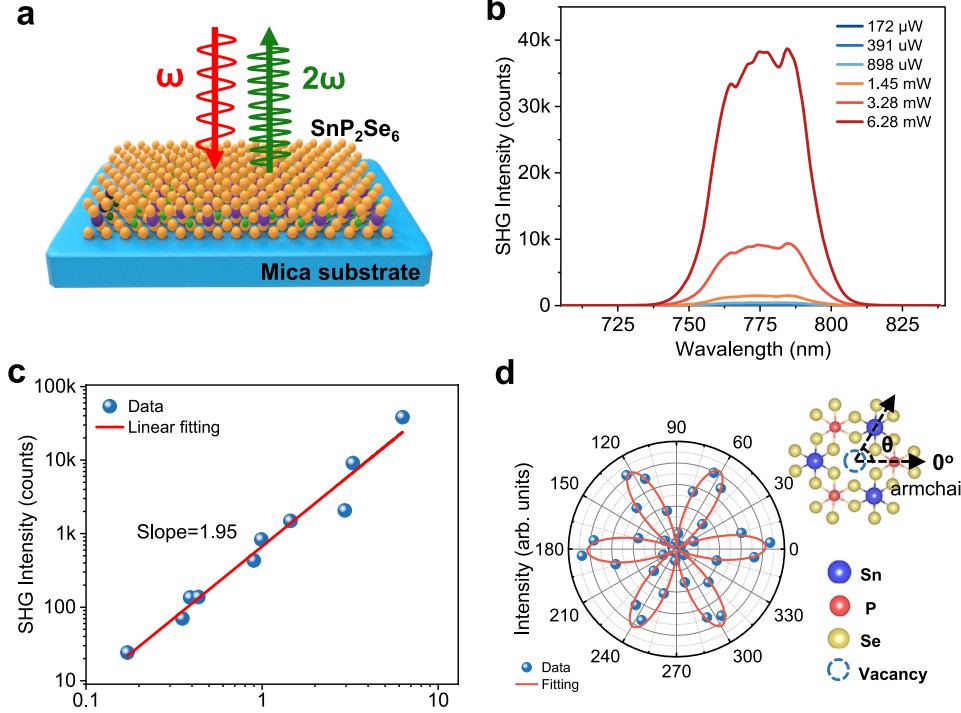

**Fig. 2 | Second harmonic generation (SHG) characterization of layered SnP₂Se₆.** **a** Schematic of SHG measurement. The symbols $\omega$ and $2\omega$ in the diagram represent the frequencies of the incident light and the SHG light, respectively. **b** Power-dependent SHG spectra of SnP₂Se₆ nanosheets under a 1550 nm laser. **c** The excitation power dependence of SHG intensity in logarithmic coordinate. The red line was obtained by linear fitting. **d** Polarization-dependent SHG spectra of SnP₂Se₆ nanosheets and corresponding SnP₂Se₆ crystal model, along with its polarization angle (The armchair direction is defined as 0°). The SHG intensity can be fitted with $I = I_0 \sin^2 (3\theta)$ in parallel configuration, where $I$ is the SHG intensity at angle $\theta$, and $I_0$ is the maximum SHG intensity.

the range from 8 L to bulk, thus leading to the deviation from quadratic relationship even with a downward trend[28]. We did not take 3R-MoTe₂ for comparison, since the current synthesis techniques usually yield samples with small size[29], which causes immense hardship to be integrated into SiN photonics for on-chip electronic-photonic co-design.

To clarify the origin of SHG activity in SnP₂Se₆ crystals, we conducted cross-sectional AC-STEM measurements to reveal the stacking configuration at the atomic scale. In sharp contrast with TMDCs showing 2H stacking or 3 R stacking configurations, we noted that there is no period structure along the direction normal to the (001) plane of monolayer SnP₂Se₆ (Fig. 3c). Similar case has been reported in spiral WS₂ nanosheets, which is associated with the distortion caused by the screw growth[19]. However, the absence of periodic stacking in 2D SnP₂Se₆ is entirely unrelated to the strain, and it can be well explained by the weak interaction between layers. Figure 3d compares the atomic structures of 2H-MoTe₂ and SnP₂Se₆. As previously reported, the disappear of SHG intensity in even-layered 2H-MoTe₂ is interpreted by the restoration of inversion symmetry[27]. In SnP₂Se₆, the complete absence of period stacking along the [0001] direction indicates that the non-centrosymmetric stacking configuration can be well maintained with layer increasing. Effective coupling enhancement in thick sample would cause the superposition of SHG signals, thus contributing to the excellent SHG efficiency without layer-number limitation[30,31]. According to the following formula[32]:

$$\chi^{(2)} = \frac{\varepsilon_0^{1/2} c^{1/2} \lambda A^{1/2}}{8^{1/2} \pi} \cdot \frac{1}{P_\omega} \cdot \frac{1}{\mathrm{d}} \cdot P_{2\omega}^{1/2} \cdot n_\omega n_{2\omega}^{1/2} \tag{1}$$

where $P_\omega$ and $P_{2\omega}$ represent the excitation laser power and SHG power, $d$ is the thickness of sample, $\varepsilon_0$ is the dielectric constant, $c$ is the speed of light in vacuum, $A$ is the area of incident laser spot, $n_\omega$ and $n_{2\omega}$ denote the linear refractive indices of the sample at the fundamental and SH

frequencies, respectively. We can obtain a high $\chi^{(2)}$ -1.3 × 10⁻⁹ m·V⁻¹ for 2D SnP₂Se₆ under 1550 nm wavelength. This value is comparable to that of 2H-MoTe₂ (-2.5 × 10⁻⁹ m·V⁻¹)[27] and among the highest values of the reported 2D semiconductors (Fig. 3e). More importantly, the considerably high SHG susceptibility and odd-even layer-independence of SHG response address the dilemma of sufficiently high conversion efficiency and restored inversion symmetry in even-layered samples, and it is of crucial importance for the practical device applications.

## Electrical and optoelectronic properties of 2D SnP₂Se₆

2D layered materials applied for monolithic electronic-photonic integration shall be of excellent semiconducting properties. To evaluate the electrical properties of 2D SnP₂Se₆, field-effect transistors (FETs) with the back-gate configuration were fabricated (Supplementary Fig. 10). The switching characteristic obtained from a device with a 10-nm-thick channel demonstrates a typical electron-dominated transport behavior. Thickness-dependent field-effect mobility and ON-OFF ratio, two key metrics of device performance, were evaluated based on the data from more than forty devices. By extracting the data from the linear region of transfer curve, we can obtain a field-effect mobility ($\mu_{EF}$) of 15 cm² V⁻¹ s⁻¹ for a 20-nm-thick sample. The ON-OFF ratio of device drops monotonically from ~10⁶ to ~10² with the increasing of thickness (Supplementary Fig. 11). The excellent performance of SnP₂Se₆ FETs in terms of high electron mobility and ON-OFF ratio suggests its potential applications for logic electronics. Temperature-dependent transport characteristics of the device were shown in Supplementary Fig. 12. Attributed to the electron-phonon scattering[33], $\mu_{EF}$ increases as the temperature decreases from 300 to 200 K. It approximately follows the relation $\mu_{EF} = T^{-\gamma}$, where $\gamma$ represents the phonon damping factor. By fitting this curve, the value of $\gamma$ is estimated to be 1.3, which is smaller than that of MoS₂[34] and B-P[35]. Note that $\mu_{EF}$ decreases with the further dropping down of temperature to 7 K, indicating the scattering from

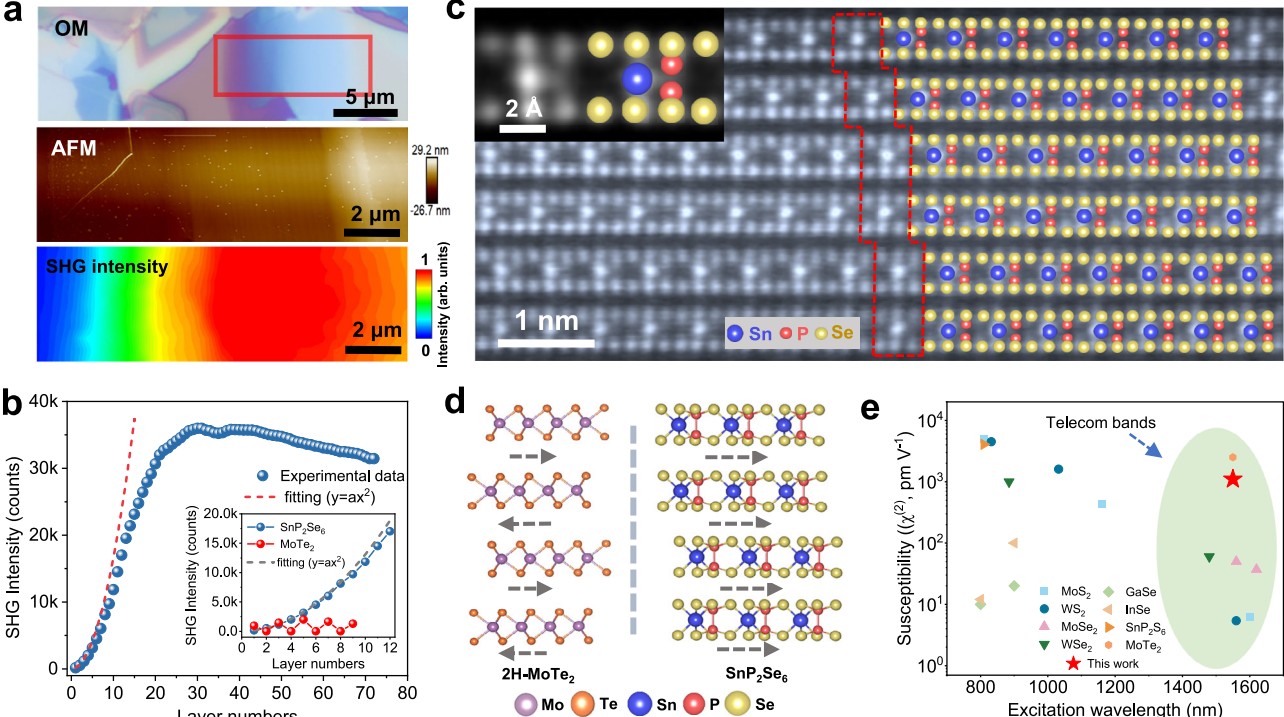

**Fig. 3 | Thickness-dependent SHG characterization of layered SnP$_2$Se$_6$. a** Optical microscopy (OM) image, AFM topography image and corresponding SHG intensity mapping of SnP$_2$Se$_6$ nanosheet with different thicknesses. **b** Thickness-dependent SHG intensities of SnP$_2$Se$_6$. Inset image compares the SHG response of SnP$_2$Se$_6$ and 2H-MoTe$_2$. The dashed lines (including the inset) represent the curves fitted using the formula $y = ax^2$, which depict the relationship between the SHG intensity and the layer numbers in the ideal state. **c** Cross-sectional AC HAADF-STEM image of SnP$_2$Se$_6$ nanosheet, together with the corresponding side-view atomic model. The red dotted outline reflects the interlayer stacking form of SnP$_2$Se$_6$. **d** Comparison of interlayer stacking configurations between 2H-MoTe$_2$ and SnP$_2$Se$_6$. The dashed arrows represent the interlayer stacking direction of the crystals. **e** Comparison of second-order susceptibility ($\chi^{(2)}$) between SnP$_2$Se$_6$ and other reported 2D materials[18,27,32,45].

charged impurities dominates the mobility behavior of 2D SnP$_2$Se$_6$[36]. By optimizing the device configuration, such as using a top-gated structure and high-$k$ gate dielectric, we can expect the significant improvement of device performance since the charge impurities at heterointerface can be effectively screened.

The excellent visible-light absorption properties indicate 2D SnP$_2$Se$_6$ might be an ideal candidate for light detection, thus the performance of SnP$_2$Se$_6$ phototransistor was investigated as a laser is scanned over the channel of device. Figure 4a plots the transfer curves of a SnP$_2$Se$_6$ phototransistor under dark and 700 nm light illumination with varying optical power densities. The inset image illustrates 3D-view AFM tomography image of the device. Significant photocurrent is generated as the gate voltage ($V_g$) scans from −50 to 50 V. At $V_g = -45$ V, the device demonstrates a high ON-OFF ratio, reaching up to $10^4$ under an illumination intensity of 10 mW cm$^{-2}$. The broadband photoresponse (300 – 900 nm) enables the device to realize optical detection of the full visible wavelength (Fig. 4b and Supplementary Fig. 13). Laser beam scanning across the device (power of 75 μW, wavelength of 700 nm) yields a spatial mapping of the photoresponse (Inset in Fig. 4b). The photocurrent signals mainly occurred in the channel area, which is indicative of the photoconductance-dominated the generation of photocurrent.

Photoresponsivity ($R_\lambda$) and detectivity ($D^*$) are the two important parameters for photodetectors, and they can be defined by formulars:

$$R_\lambda = I_{\text{ph}}/PS \tag{2}$$

$$D^* = RS^{1/2}/\left(2qI_{\text{d}}\right)^{1/2} \tag{3}$$

where $I_{\text{ph}}$, $I_{\text{d}}$, $P$, and $S$ represent the photocurrent, dark current, incident power, and effective illuminated area, respectively. The device shows gate-tunable $R_\lambda$ under a light power density of 0.2 mW cm$^{-2}$ and a bias of 2 V, yielding a maximum value up to $10^3$ A W$^{-1}$, and the $D^*$ reaches its maximum value of $5.1 \times 10^{10}$ Jones at a gate of −38 V (Fig. 4c). In addition, the dynamic response demonstrates good cycling performance and fast operation speed for the devices. The rise and decay time constants are figured out to be about 412 and 910 μs, respectively (Fig. 4d, e and Supplementary Fig. 14). The high photoresponsivity and speed are among the best value in the reported phototransistors based on 2D layered semiconductors[37]. We also noted that 2D SnP$_2$Se$_6$ exhibit satisfied stability as stored in air condition. The surface morphology of SnP$_2$Se$_6$ channel can be well maintained. Insignificant degradation in electrical performance was observed after the device being exposed in air for one week without any encapsulation (Fig. 4f). The high electron mobility, moderate band gap, great ambient stability and pronounced absorption in visible region indicate 2D SnP$_2$Se$_6$ could be a promising candidate for application in high-performance electronic and optoelectronic devices.

## On-chip integration of optical modulation and detection

Benefiting from the odd-even layer-independent SHG activity and high SHG susceptibility at telecom wavelength, multilayer SnP$_2$Se$_6$ with sufficient thickness for high-efficient frequency conversion can be integrated into SiN photonic platform, thus enabling the design of on-chip optical modulator[14]. Compared with the free-space interaction mode, a remarkable enhancement of SHG process can be expected by integrating 2D NLO materials into photonic microring resonator due to strong light-matter coupling effect[7]. Figure 5a displays the OM image of a SiN photonic structure for on-chip SHG, in which two bus-

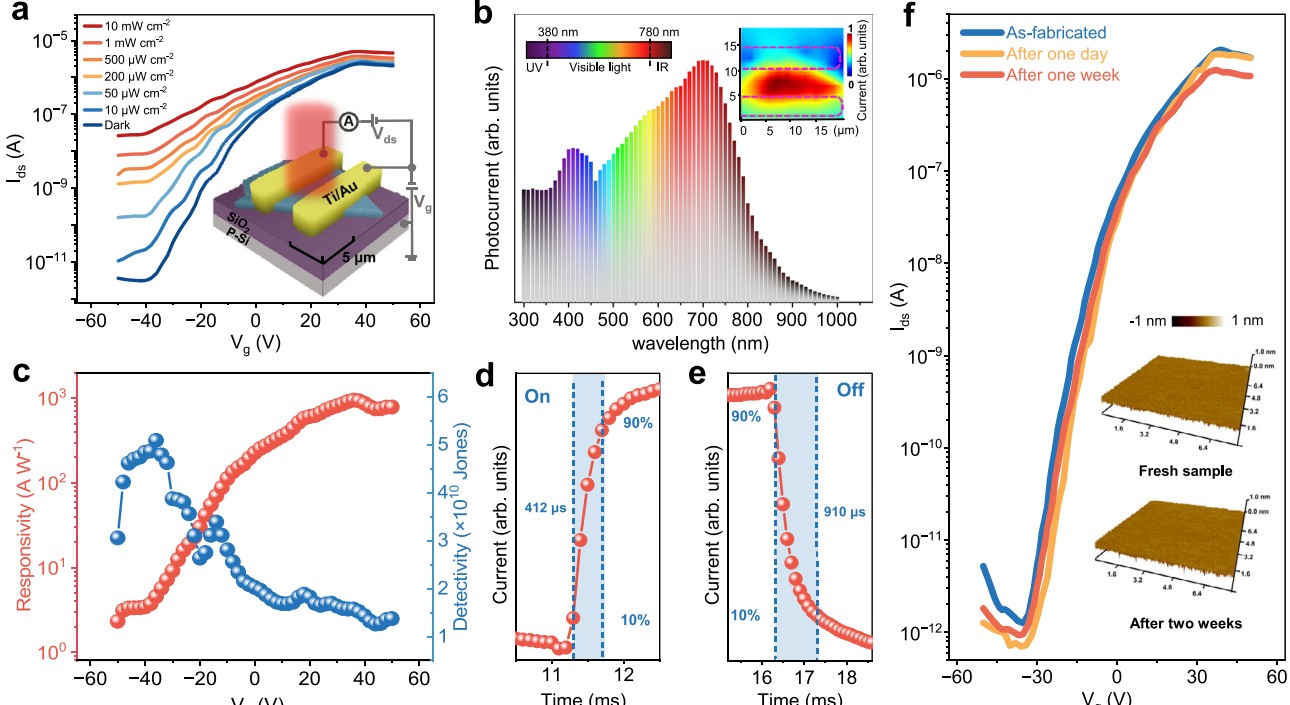

**Fig. 4 | Electrical and optoelectronic properties of SnP$_2$Se$_6$ phototransistors.**
**a** Typical transfer curves of SnP$_2$Se$_6$-based phototransistor under dark and 700 nm light illumination with different optical power densities. The $I_{ds}$, $V_{ds}$ and $V_g$ represent the source-drain current, source-drain bias voltage and gate voltage, respectively. Channel length and width of the device are 5 μm and 10 μm, respectively. Inset image shows the schematics of the device. **b** Photoresponse of device as a function of wavelength ranging from 300–1000 nm at $V_{ds} = 1$ V with optical power density of 20 mW cm$^{-2}$. The inset displays photocurrent map under 700 nm illumination. The device and electrodes are indicated by dashed lines. **c** Gate voltage-dependent responsivity and detectivity obtained from Fig. 3a. **d, e** Rise and decay curves measured using an oscilloscope. **f** Transfer curves of a device with stability measured for up to two weeks. Inset image plots the corresponding surface morphology of device channel measured by AFM.

waveguides are coupled with a SiN microring resonator. A fundamental pump laser (~1560 nm) can be coupled into bus-waveguide ($WG_{pump}$) from free space through a grating coupler. The generated SHG signals is coupled out from microring to sub-waveguide ($WG_{SHG}$), and finally collected by a fiber spectrometer on top of another grating coupler. To satisfy the phase-matching condition, the structure of SiN microring resonator is carefully designed. As shown in Supplementary Fig. 15, both effective refractive indices of the fundamental mode and SHG mode increases with the strip width of microring. A cross point at $w = 1.24$ μm indicates that the phase matching condition can be well satisfied for the SHG process in this configuration.

A 10 nm-thick SnP$_2$Se$_6$ nanosheet with hundreds of micrometers in size is then transferred onto the device, covering the whole region of SiN microring resonator (Fig. 5b). The on-resonance light would circulate in the microring and further evanescently interact with SnP$_2$Se$_6$ nanosheet to generate SHG signals. The inset images in Fig. 5b depict the mode profiles of optical transmission at different locations of device. Note that the fundamental guiding mode TE$_0$ at 1560 nm (with the polarization parallel to the waveguide plane) would dominate the light propagation in the $WG_{pump}$ and microring due to the low transmission loss, while it would change to TM$_2$ mode (with the polarization normal to the waveguide plane) after the SHG process from 1560 to 780 nm (Supplementary Fig. 16). With the assistance of a mode converter, the TM$_2$ mode of microring is further converted to fundamental TM$_0$ mode in $WG_{pump}$ (Supplementary Fig. 17). The experimental implementation of SHG from SnP$_2$Se$_6$ – SiN microring resonator is based on a home-built optical measuring system, as shown in Supplementary Fig. 18.

Supplementary Fig. 19 shows the transmission spectra of SiN microring after SnP$_2$Se$_6$ integration. The collected transmission spectrum periodically dips with a free spectral range of 9 nm, which is indicative of the efficient coupling between guiding mode of $WG_{pump}$ and resonance modes in the microring. By fitting the resonance dips with Lorentzian line shapes, the Q factor of the resonance modes is calculated to be 1000. It should be noted that the resonance dips would red-shift by 1.7 nm after SnP$_2$Se$_6$ overlapping, which is attributed to the perturbation of resonance modes arised from the high refractive index of SnP$_2$Se$_6$[7]. As expected, strong optical signals around 780 nm were collected from the output grating coupler by exciting SnP$_2$Se$_6$ nanosheet from the SiN microring resonator. The peak intensity scales quadratically with the excitation power, further confirming the observed signals are from the SHG (Fig. 5c). We then estimate and optimize the SHG conversion efficiency of SnP$_2$Se$_6$ nanosheet integrated on SiN microring resonator (Supplementary Figs. 20–22). For an incident optical power of 10 mW, the effective power ($P_{pump}$) coupled into $WG_{pump}$ is estimated to be 356 μW by normalization of the coupling loss. Though only a small portion of the incident light is launched into the sub-waveguide due to the coupling loss, the grating couplers design can be further optimized to improve the coupling efficiency. The up-converted SHG signals was finally collected from the output grating coupler, with an estimated power of $P_{SHG} = 55$ nW. Therefore, the conversion efficiency ($\eta$) of SHG can be calculated through the following equation[7,38]:

$$\eta = P_{SHG}/P_{pump}^2 \times 100\% \qquad (4)$$

The maximum value of 43.2%W$^{-1}$ can be obtained for a sample with thickness about 21 nm, which is reasonably satisfying for our experimental setup and notable in previously reported TMDCs-integrated photonic structures[11,39].

Photodetector operated at telecom wavelength is the key device of electronic-photonic integrated circuits. However, 2D SnP$_2$Se$_6$ with a

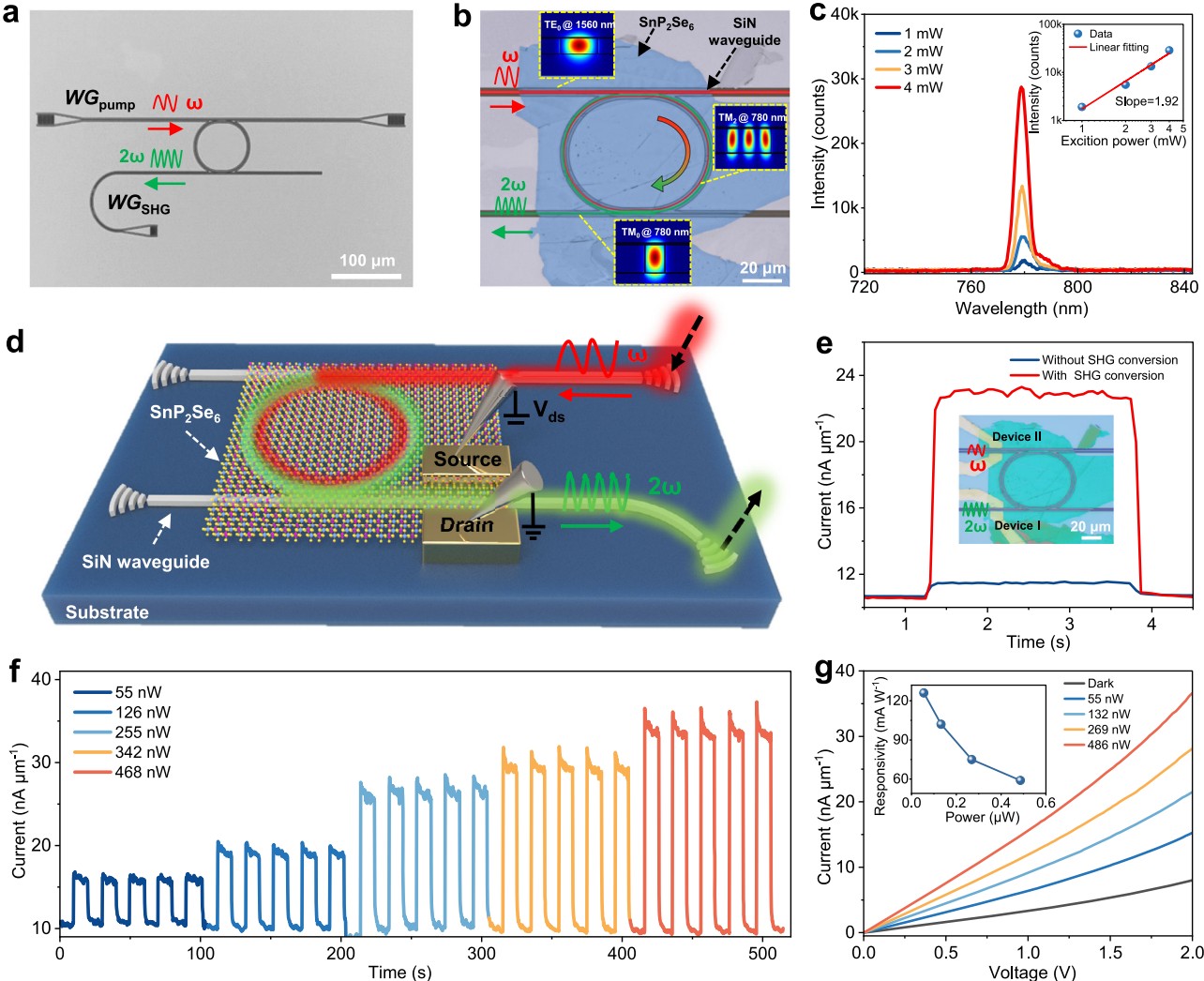

**Fig. 5 | 2D SnP₂Se₆ for on-chip optical signal modulation and detection. a** OM image of the fabricated SiN photonic structure, in which two bus-waveguides are coupled with a SiN microring resonator. $WG_{pump}$ and $WG_{SHG}$ represent the grating couplers used for coupling the pump light and SHG, respectively. **b** OM image of an on-chip optical modulator based on SiN/SnP₂Se₆ hybrid structure. Inset images are the simulated mode profiles, including the pump source (TE₀ @ 1560 nm) in the input bus-waveguide, the SHG signals (TM₂ @ 780 nm) in the microring resonator and the SHG signals (TM₀ @ 780 nm) in the output bus-waveguide. **c** Emission spectra collected from the grating coupler after SHG process. Inset image shows

the dependence of SHG intensity on the excitation power. The red line in the inset was obtained by linear fitting. **d** A schematic of the SHG-assisted SnP₂Se₆ photo-detector. **e** Photoresponse of the devices with and without SnP₂Se₆ integration for frequency pre-modulation. Inset shows the OM image of photodetector. Device I and device II are fabricated on $WG_{SHG}$ and $WG_{pump}$, respectively. **f** Time-resolved photoresponse of devices under different input light powers. **g** $I$–$V$ curves of the device as a function of input light power. Inset is the responsivity of the SHG-assisted SnP₂Se₆ photodetector as a function of illuminated optical power.

band gap in the range from 1.75 − 1.90 eV is expected to exhibit weak photoresponse due to the poor sub-bandgap absorption in telecom band. Combining the merits of high SHG efficiency and strong photoresponse in visible wavelengths of SnP₂Se₆, we designed a monolithic on-chip photodetector, which operates based on the upconversion of optical signals from telecom band to visible band. Figure 5d and inset image in Fig. 5e plots the schematic diagram and OM image of device, respectively. The photodetector is fabricated at the end of SnP₂Se₆ nanosheet along the wave transmission path, with Ti/Au (10/80 nm) electrodes patterned on the opposite sides of $WG_{SHG}$. On-chip frequency conversion from 1560 to 780 nm is first implemented through the SnP₂Se₆-SiN microring resonator. The converted 780 nm light propagating along $WG_{SHG}$ can be overlapped evanescently with the absorbing SnP₂Se₆ nanosheet in the active section of the detector. As shown in Fig. 5e, device I is fabricated on top of $WG_{SHG}$. Relying on the upconversion of light, a remarkable photoresponse can be detected. To eliminate the influence of intrinsic optical absorption at 1560 nm, device II fabricated on $WG_{pump}$ without

light upconversion was also measured for comparison. We can see that the photocurrent is negligible under illumination with identical exciting power, which is several orders of magnitude smaller than that measured in device with wavelength conversion. The result strongly confirms the decisive role of up-converted visible light on the photo-generation in device.

Figure 5f plots the time-resolved photoresponse of devices under different input light powers. A pronounced current enhancement was detected when light is coupled in. Figure 5g exhibits the $I$-$V$ curves under different input light powers, and the variation in responsivity with dependence on input power was also calculated (inset in Fig. 5g). By normalizing the coupling loss and waveguide propagation loss, a reasonable overall responsivity ($R$) of 126 mA W⁻¹ is obtained for the integrated device, which has taken both SHG conversion and photo-detection into account. The estimated $R$ is comparable with the current state-of-the-art photodetectors based on InGaAs[40], Ge[41] and narrow-band-gap 2D layered materials (*e.g.*, B-P[42], Bi₂O₂Se[43] and graphene[44]), indicating its potentials for on-chip telecom-band

photodetection. We expect that by optimizing device configuration, the conversion efficiency of SHG and photoresponsivity of device can be further improved. More importantly, the monolithically integrated functions, including the frequency conversion and photodetection of optical signals at telecom wavelength, are first implemented simultaneously in a single-unit device based on the $SnP_2Se_6$ NLO semiconductor. This work provides opportunities for the development of multifunction integration in next-generation on-chip EPICs.

## Discussion
In summary, using a space-confined chemical vapor transport strategy, we first synthesized 2D $SnP_2Se_6$ layered semiconductor with strong second-order NLO properties, and demonstrated its applications for monolithic on-chip electronic-photonic integration. Owing to its unique inversion symmetry and interlayer stacking configuration, 2D $SnP_2Se_6$ exhibits odd-even layer-independent SHG response with a high susceptibility of $1.3 \times 10^{-9}$ m·V$^{-1}$ under 1550 nm excitation wavelength. 2D $SnP_2Se_6$ phototransistor delivers a high responsivity of $10^3$ A W$^{-1}$ and fast response rate (412/910 µs) under 700 nm illumination. The SiN/$SnP_2Se_6$ hybrid photonic device reveals a strong on-chip SHG process with conversion efficiency of 43.2%W$^{-1}$. Meanwhile, combining the excellent SHG activity and optoelectronic properties, a prototype on-chip telecom-band photodetector relying on the upconversion of optical signals from 1560 nm to 780 nm was first experimentally demonstrated. The implementation of monolithic integrated multifunctions including on-chip optical modulation and photodetection indicates 2D $SnP_2Se_6$ is promising to fulfill the requirement of chip-level electronic-photonic co-design for EPICs.

## Methods
### Synthesis of 2D $SnP_2Se_6$ crystals
The schematic diagram of the self-limited epitaxy strategy to grow 2D $SnP_2Se_6$ crystals is shown in Supplementary Fig. 1. Stoichiometric amounts of Sn powders (Adamas, 99.9%), P powders (Adamas, 99.9%) and Se powders (Adamas, 99.99%) are sealed at the end of a horizontally-placed quartz tube with pressure less than 10 mbar. Meanwhile, serving as the substrate for self-limited epitaxy growth, fresh-exfoliated fluorophlogopite ($KMg_3(AlSi_3O_{10})F_2$) sheets (Taiyuan Fluorophlogopite Co. Ltd., Changchun, China) were peeled off, and then put into the other end of the vacuum quartz tube. The vacuum quartz tube was placed in a tube furnace and heated to 580 °C within 3 h. After 6 h, the furnace was slowly cooled to room temperature at a rate of 0.5 °C min$^{-1}$. Ultrathin $SnP_2Se_6$ nanosheets were successfully synthesized in the sandwiched mica sheets.

### DFT calculations
The first-principal calculations were performed by using the Vienna ab initio simulation packages (VASP). We employed the Perdew-Burke-Ernzerhof (PBE) parametrization of the generalized gradient approximation for the exchange-correlation energy. The force on each atom is less than 0.02 eV Å$^{-1}$ and the energy tolerance is $10^{-5}$ eV respectively for structure relaxation. The energy cut-off of the plane wave basis is chosen to be 600 eV. A gamma-centered $6 \times 6 \times 1$ Monkhorst-Pack k-point mesh was applied for the k-point samples in the Brillouin zone.

### Materials characterization
The morphology of as-prepared $SnP_2Se_6$ nanosheets was characterized by optical microscope (Axioscope 5, Zeiss) and AFM (Dimension Icon, Bruker). Raman spectroscopy, micro-UV-visible absorption spectroscopy, and SHG spectroscopy were carried out by Metatest ScanPro Laser Scanning System (ScanPro Advance, Metatest). The chemical valence and bonding states of $SnP_2Se_6$ nanosheets was

characterized by XPS (Thermo Scientific NEXSA, Thermo Fisher). HAADF-STEM images and EDS mapping were recorded by using a double $C_s$-corrected FEI-Themis microscope operated with an acceleration voltage of 200 kV. The cross section of $SnP_2Se_6$ nanosheet was obtained by focused ion beam (Scios, FEI).

### Device fabrication and measurement of $SnP_2Se_6$ phototransistor
$SnP_2Se_6$ nanosheets were transferred from mica substrates to $SiO_2$/Si substrates (300 nm $SiO_2$) with the assistance of the polymethyl methacrylate (PMMA). The $SnP_2Se_6$ FETs were then fabricated using laser direct writing (Microlab 4A100, SVG Optronics. Co., Ltd.) followed by electron-beam evaporation (TEMD500, Beijing Technol Science Co., Ltd.). The electrical performance of $SnP_2Se_6$ FETs was measured in a probe station (CRX-6.5 K, Lake Shore Cryotronics, Inc.) with the assistance of a semiconductor analyzer (4200A-SCS, Keithley). The optoelectrical measurement of $SnP_2Se_6$ phototransistor was carried out on the Metatest ScanPro Laser Scanning System (ScanPro Advance, Metatest) with a wavelength-adjustable xenon lamp (280–1000 nm) as the light source.

### Fabrication of $SnP_2Se_6$/SiN hybrid device
To avoid the light absorption at visible wavelength, the waveguide and microring resonator were fabricated on a 300 nm-thick SiN slab, which are deposited on Si substrate coated with 3 µm-thick buried oxide layer. The widths of two coupling bus waveguides are different, which are designed to accomplish the coupling in of the fundamental pump laser (at 1560 nm) and coupling out of the SHG signals (at 780 nm). A 200 nm-thick $SiO_2$ cladding was deposited by the electron beam evaporation (SKE-A-75) above SiN slab, and the devices were patterned by the electron beam lithography (Raith eLINE) using positive photoresist (ZEP 520 A). The $SiO_2$ layer were then fully etched to a depth of 200 nm, and the patterned layer of $SiO_2$ was utilized as the hard mask for the following SiN etching. Both two steps of etching are carried out with the assistance of inductively coupled plasma (ICP) dry etching. The fabrication process is schematically plotted in Supplementary Fig. 23. After that, the $SnP_2Se_6$ nanosheets were transferred and covered on the surface of the SiN microring resonator. $SnP_2Se_6$ photodetector was fabricated along the wave transmission path with Ti/Au (10/80 nm) electrodes patterned on the opposite sides of the waveguide.

### Performance measurement of $SnP_2Se_6$/SiN hybrid device
The home-built optical measuring system consists of a tunable laser (Santec TSL-710) at C band, a polarization controller, a fiber-chip coupling stage and a benchtop power meter (MPM210). The single-mode fibers are used to connect the I/O grating couplers. The polarization controller was used to excite the $TE_0$ guiding mode in the SiN waveguide. The transmission spectra are acquired by measuring the output power and scanning the wavelength from 1500 to 1600 nm with a step of 1 pm. The conversion efficiency of SHG is obtained by measuring the effective power coupled into the SiN waveguide ($P_{pump}$) and effective power of SHG ($P_{SHG}$).

## Data availability
The data generated in this study are provided in Source data. Extra data that support the findings of this study are available from the corresponding authors upon request. Source data are provided with this paper.

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

## Acknowledgements

This work was supported by National Natural Science Foundation of China (Nos. 52102161 and 61875049), Natural Science Foundation of Guangdong Province (Nos. 2021A1515012423 and 2022B1515020057), Shenzhen Science and Technology Program (Nos. GXWD20201230155427003-20200805161204001, RCBS20200714114911270, RCYX20210609103707009, RCJC20210706091950025, and RCYX2022100809291204), and Research Grant Council of Hong Kong (15205318).

## Author contributions

C.-Y.Z. and Z.Z. contributed equally to the work. J.-K.Q., K.X., Y.C. and C.-Y.X. conceived the idea and proposed the research. C.-Y.Z. and P.-Y.H. performed the growth experiments and analyzed the experimental data. C.-Y.Z., Z.W., and Y.Z. performed device fabrication and analyzed the experimental data. Z.Z., Y.L., and K.X. conducted and supervised the on-chip optical transimission measurement. C.-Y.Z., P.M. and L.Z. performed and analyzed the SHG measurement. C.W. and Y.C. performed and supervised the DFT calculations. J.-K.Q., C.-Y.Z., K.X., Y.C. and C.-Y.X. co-wrote the manuscript.

## Competing interests

The authors declare no competing interests.
