## [Peer Review File · Nature Communications]

Two-dimensional semiconducting SnP₂Se₆ with giant second-harmonic-generation for monolithic on-chip electronic-photonic integrationEditorial Note: This manuscript has been previously reviewed at another journal that is not operating a transparent peer review scheme. This document only contains reviewer comments and rebuttal letters for versions considered at Nature Communications.

REVIEWER COMMENTS

Reviewer #1 (Remarks to the Author):

The authors have replied to the comments of Referees 1 and 2, but I do not find the response letter fully satisfactory. I would in principle support the publication of the manuscript in Nature Communications if the following points are properly addressed.

1. I am not at all convinced by the arguments against 3R-phase TMDs. For instance ref.20 (Towards compact phase-matched and waveguided nonlinear optics in atomically layered semiconductors, now on Nature Photonics) uses 3R phase MoS₂ and shows similar or even better performances compared to the one reported in this manuscript, also in a similar wavelength (telecom) range. 3R MoS₂ is thermodynamically stable and can be produced on large scale.
2. The authors write "Note that the 3R-stacked TMDCs are considered as metastable phase, which usually exhibit unsatisfied electrical and optoelectronic properties, and they can only be obtained through mechanical exfoliation or chemical vapor deposition under harsh conditions [21]". In my opinion this is not the message of ref.21. On the contrary, ref. 21 states that 3R MoS₂ is thermodynamically stable and can be produced by different methods, including large scale CVT.
3. The discussion on the linear scaling with number of layers is not convincing. The figure reported in the response letter (Figure R1 from ref. 1 of the SI) starts from a thickness of > 7nm, where effects such as re-absorption might play an important role. Instead Fig.2e of this manuscript (y-axis still missing, please specify if this is linear or log. scale) is in the range 1-18 layers. In the range <5-10 layers, re-absorption should be negligible and the scaling with number of layers should be quadratic (see e.g. also <https://doi.org/10.1038/lsa.2016.131>). I find this linear dependence really surprising, and I strongly recommend to further investigate this point.
4. In the response to Referee 2, the authors comment that they can't obtain large area monolayer samples. What about 2 and 3 layers, as additionally requested by Referee 2? This would still allow to compare experimental results and theory.

Reviewer #2 (Remarks to the Author):

The authors addressed most of the comments. I hope the authors could further clarify two remaining issues.

(1) I don't think the authors addressed my question on the contact resistance. It seems to me that the device does not have an ideal Ohmic contact. It will be great if the authors can provide the estimation of the resistance.

(1) The authors did not address my comment on the efficiency. Simply, following the equation 4, P(SHG) is 46 nW, while P(pump) is 338 uW. I do not understand how the 40.3% was obtained. Further, Could the author provide a theoretical limit of the efficiency as a function of the layer thickness?

Responses to the reviewers' comments (NCOMMS-22-43928)

Reviewer #1 (Remarks to the Author):

The authors have replied to the comments of Referees 1 and 2, but I do not find the response letter fully satisfactory. I would in principle support the publication of the manuscript in Nature Communications if the following points are properly addressed.

1. I am not at all convinced by the arguments against 3R-phase TMDs. For instance, ref.20 (Towards compact phase-matched and waveguided nonlinear optics in atomically layered semiconductors, now on Nature Photonics) uses 3R phase MoS₂ and shows similar or even better performances compared to the one reported in this manuscript, also in a similar wavelength (telecom) range. 3R MoS₂ is thermodynamically stable and can be produced on large scale.

Reply: Thanks for your kind suggestions. We have revised the corresponding description as shown below:

"The inversion asymmetry can be well maintained with layer increasing in few-layer TMDCs stacked in 3R configuration, which contributes to giant SHG activity for on-chip nonlinear optical devices"

We also cited the related papers about 3R-TMDCs as references:

*"[22] Dong, Y. et al. Giant bulk piezophotovoltaic effect in 3R-MoS₂. Nat. Nanotechnol. **18**, 36–41 (2023).*

*[23] Trovatiello, C. et al. Optical parametric amplification by monolayer transition metal dichalcogenides. Nat. Photon. **15**, 6-10 (2021).*

[24] Xu, X. *et al.* Towards compact phase-matched and waveguided nonlinear optics in atomically layered semiconductors. *Nat. Photon.* **16**, 698-706 (2022). "

2. *The authors write "Note that the 3R-stacked TMDCs are considered as metastable phase, which usually exhibit unsatisfied electrical and optoelectronic properties, and they can only be obtained through mechanical exfoliation or chemical vapor deposition under harsh conditions [21]". In my opinion this is not the message of ref.21. On the contrary, ref. 21 states that 3R MoS₂ is thermodynamically stable and can be produced by different methods, including large scale CVT.*

Reply: Thanks for reviewer's comments. We have deleted corresponding discussion as reviewer suggested.

3. *The discussion on the linear scaling with number of layers is not convincing. The figure reported in the response letter (Figure R1 from ref. 1 of the SI) starts from a thickness of > 7nm, where effects such as re-absorption might play an important role. Instead Fig.2e of this manuscript (y-axis still missing, please specify if this is linear or log. scale) is in the range 1-18 layers. In the range <5-10 layers, re-absorption should be negligible and the scaling with number of layers should be quadratic (see e.g. also <https://doi.org/10.1038/lssa.2016.131>). I find this linear dependence really surprising, and I strongly recommend to further investigate this point.*

Reply: Thanks for reviewer's comments. We paid high attention to the reviewer's concerns about the thickness dependent SHG intensity and systematically conducted related characterization. The experimental results are quite different from those previously discussed.

Fig R1a exhibits the OM image (top), AFM topography image (middle) and

corresponding SHG intensity mapping (bottom) of SnP₂Se₆ nanosheet with different thicknesses, and **Fig R1b** displays the relationship between the SHG intensity and layer numbers. One can see that the SHG intensity would increase with the layer number (< 7 layers), revealing a near-quadratic dependence. By comparison, in the range from 8 layer to bulk, the influence of light re-absorption is being strengthened gradually, leading to the deviation from the quadratic relationship even with a downward trend^{1,2}. This phenomenon well matches with the reviewer’s opinion. Therefore, we revised the inaccurate description in maintext.

Fig. R1 Thickness-dependent SHG of SnP₂Se₆ nanosheets. **a** OM image (top), AFM topography image (middle) and corresponding SHG intensity mapping (bottom) of SnP₂Se₆ nanosheet with different thicknesses. **b** Thickness-dependent SHG intensities of SnP₂Se₆.

Fig. R1 has been updated as part of **Fig. 2**, and the revised discussion is shown as below:

“In contrast, SnP₂Se₆ nanosheet with few layers (< 7L) demonstrates quadratically increased SH response, and the collected SHG peak intensity for multilayer SnP₂Se₆ nanosheet exceeds that of 2H-MoTe₂ with layer number increasing”

“It should be noted that the influence of light re-absorption is being strengthened gradually in the range from 8 L to bulk, thus leading to the deviation from quadratic

relationship even with a downward trend.”

We also cited related paper reviewer mentioned as reference:

“ [28] Zhao, M. *et al.* Atomically phase-matched second-harmonic generation in a 2D crystal. *Light: Sci. Appl.* **5**, e16131-e16131 (2016).”

4. In the response to Referee 2, the authors comment that they can't obtain large area monolayer samples. What about 2 and 3 layers, as additionally requested by Referee 2? This would still allow to compare experimental results and theory.

Reply: Thanks very much for the reviewer's comments. We have successfully obtained ultrathin SnP₂Se₆ nanosheets by mechanical exfoliation and obtained the micro-UV-visible absorption spectra of SnP₂Se₆ nanosheets with 2L, 3L, 4L, 5L, and 16L, respectively (**Fig. R2**). The value of the band gap for 2L-SnP₂Se₆ nanosheet is estimated to be 1.93 eV from the Tauc curve, and it would drop to 1.75 eV as the layer increases to 16 L.

Fig. R2 Thickness-dependent micro-UV-visible absorption spectra of SnP₂Se₆ nanosheets. Inset image exhibits optical band gap identified by the micro-UV-visible absorption

spectroscopy.

Fig. R2 has been updated as part of **Fig. 1**, and the related discussion has been updated as below:

*“The value of the band gap for 2L SnP₂Se₆ nanosheet is estimated to be 1.93 eV from the Tauc curve (inset image in **Fig. 1**), and it would drop to 1.76 eV as the layer increases to 16L.”*

Reviewer #2 (Remarks to the Author):

The authors addressed most of the comments. I hope the authors could further clarify two remaining issues.

1. I don't think the authors addressed my question on the contact resistance. It seems to me that the device does not have an ideal Ohmic contact. It will be great if the authors can provide the estimation of the resistance.

Reply: Thanks for the reviewer's suggestions. According to the reviewer's suggestion, we have estimated the contact resistance of device. The total resistance includes two parts: channel resistance and contact resistance ($R=R_{channel}+2R_{contact}$), and it can be measured using the transfer length method. **Fig. R3a** shows the OM image of device based on 16-nm-thick SnP₂Se₆ nanosheet with channel lengths ranging from 2 to 9 μm. **Fig. R3b** shows the total resistance of devices in the on-state mode ($V_g=40$ V). By extracting the intercepted value with ordinate, we can obtain the $R_{contact}$ of 24.5 kΩ·μm. The contact resistance can be further reduced by optimizing electrode deposition process or replace the contact layer metal.

Fig. R3 Measurement of contact resistance of the device. **a** OM image and **b** total resistance of device with different channel lengths.

Fig. R3 has been updated as part of **Fig. S10**, and the related description has been added in the **Supplementary Information** (Part 5) as below:

*“We also estimated the contact resistance of device. The total resistance includes two parts: channel resistance and contact resistance ($R=R_{channel}+2R_{contact}$), and it can be measured using transfer length method. **Fig. S10e** shows the OM image of device based on 16-nm-thick SnP_2Se_6 nanosheet with channel lengths ranging from 2 to 8.7 μm . **Fig. S10f** shows the total resistance of devices in the on-state mode ($V_g=40\text{ V}$). By extracting the intercepted value with ordinate, we can obtain the $R_{contact}$ of 24.5 $\text{k}\Omega\cdot\mu\text{m}$.”*

2. The authors did not address my comment on the efficiency. Simply, following the equation 4, $P_{(SHG)}$ is 46 nW, while $P_{(pump)}$ is 338 μW . I do not understand how the 40.3% was obtained. Further, Could the author provide a theoretical limit of the efficiency as a function of the layer thickness?

Reply: Thanks for the reviewer’s suggestions. We apologies for the confusion caused be the absence of measure unit. The formula to calculate SHG conversion efficiency is $\eta_{TE0-TM2} = \frac{P_{SHG}}{P_{pump}^2} \times 100\%$, where P_{pump} is the power per unit length at the waveguide input, P_{SHG} is the power per unit length at the waveguide output^{3,4}. In our device, P_{pump} and P_{SHG} are estimated to be 338 μW and 46 nW, respectively. Therefore, the conversion efficiency of our second harmonic generation is calculated to be 40.3% /W.

According to the reviewer's suggestion, we also studied the relationship between the SHG efficiency and the thickness of SnP_2Se_6 nanosheet. The SHG efficiency for a lossless waveguide without pump depletion is given by the following expression^{3,4}:

$$\eta_{TE_0-TM_2} = \frac{P_{SHG}}{P_{FF}^2} \times 100\% = \xi_{NL}^2 L^2 \frac{\sin^2(\Delta\beta L/2)}{(\Delta\beta L/2)^2} \times 100\% \quad (1)$$

The Q value in the micro ring determines the overall length of propagation L , $\Delta\beta = 2\beta^{pump} - \beta^{SHG}$ represents the wave vector mismatch, and ξ_{NL} represents nonlinear overlap factor, which can be defined as:

$$\xi_{NL} = n^2 \chi_{SnP_2Se_6}^{(2)} \sin 3\theta \left(\frac{8\pi^2}{\epsilon_0 c \lambda_{FF}^2 n_{SH} n_{FF}^2} \right)^{\frac{1}{2}} \left(\frac{\int_{SnP_2Se_6} (E_{FF}^2 x)^2 E_{SH} y \, dx dz}{\left(\int_{all} |E_{FF}| \, dx dz \right)^2 \left(\int_{all} |E_{SH}| \, dx dz \right)} \right) \quad (2)$$

In this formular, θ is the angle formed by the guided-mode wave vector and the armchair direction of the SnP₂Se₆ crystal. However, since our device is fabricated based on a micro ring, this term can be eliminated by integral. Considering this, the equation (2) can be reduced to:

$$\xi_{NL} = n^2 \chi_{SnP_2Se_6}^{(2)} \left(\frac{8\pi^2}{\epsilon_0 c \lambda_{FF}^2 n_{SH} n_{FF}^2} \right)^{\frac{1}{2}} \left(\frac{\int_{SnP_2Se_6} (E_{FF}^2 x)^2 E_{SH} y \, dx dz}{\left(\int_{all} |E_{FF}| \, dx dz \right)^2 \left(\int_{all} |E_{SH}| \, dx dz \right)} \right) \quad (3)$$

where $\chi_{SnP_2Se_6}^{(2)}$ is the second-order nonlinear susceptibility of SnP₂Se₆, n is the layer of the material, $\lambda_{FF}=1550$ nm is the pump wavelength, n_{FF} and n_{SH} represents the effective refractive indices of the TE_0 mode at fundamental frequency and TM_2 mode at Second-harmonic frequency modes, respectively. ϵ_0 and c are the permittivity and light speed in vacuum. $\int_{SnP_2Se_6}$ and \int_{all} denote two-dimensional integration over SnP₂Se₆ and all space, respectively. $E_{FF}x$ is the x component of $E_{FF}(x, z)$, the electric field of the fundamental mode TE_0 , and $E_{SH}y$ is the y component of $E_{SH}(x, z)$, the electric field of the second-harmonic mode TM_2 . These electric field distribution data are obtained by Lumerical MODE solutions.

In fact, $n^2 \chi_{SnP_2Se_6}^{(2)}$ reflects the SHG conversion intensity of the SnP₂Se₆ with different layers. We systematically investigated the thickness-dependent SHG response

of SnP₂Se₆ nanosheet. **Fig R4a** exhibits OM image (top), AFM topography image (middle) and corresponding SHG intensity mapping (bottom) of SnP₂Se₆ nanosheet with different thicknesses, and **Fig R4b** displays the relationship between the SHG intensity and layer numbers. The conversion intensity is theoretically square with the number of layers. However, we noted that the SHG intensity reveals a near-quadratic dependence when the layer is less than 7, while the light re-absorption would lead to the deviation from the quadratic relationship, even showing a downward trend in the range from 8L to bulk. Therefore, we can obtain the experimental fact value of $n^2\chi_{\text{SnP}_2\text{Se}_6}^{(2)}$ with dependence on the layer numbers (**Fig. R4c**).

Fig. R4 The relationship between SHG conversion efficiency and material thickness. **a**, OM image (upper), AFM image (middle) and the SHG intensity (bottom) distribution map of SnP₂Se₆. **b**, Thickness-dependent SHG intensities (logarithmic scale) of SnP₂Se₆. **c**, The fact value of $n^2\chi_{\text{SnP}_2\text{Se}_6}^{(2)}$ with the layer numbers of SnP₂Se₆.

By substituting the data from **Fig. R4c** into equation (1) and equation (3), we can theoretically calculate the SHG conversion efficiency. **Fig. R5** shows the theoretical variation trend of the SHG conversion efficiency with material thickness. Different material thicknesses would result in various propagation constants, electric field distribution of incident waves, and second harmonics for the situation of fixed propagation length and waveguide width, which is the primary cause for the difference

in SHG conversion efficiency. Here, the waveguide width was set to be $1.26\ \mu\text{m}$ and the bending radius of the micro ring to $25\ \mu\text{m}$. According to **Fig. R5**, we can conclude that the number of material layers providing the optimal SHG conversion efficiency is around 64. It's important to note that this value is highly relative to the device configuration. Once the size parameter of the micro ring changes, the optimal conversion layer of the material will also change.

Fig. R5 The variation of SHG conversion efficiency with the thickness of the transferred SnP_2Se_6 on the micro ring device, as measured by logarithmic scale.

To further prove the above conclusion, we also supplemented the experimental data of SHG efficiency in micro ring structures covered with different thicknesses of SnP_2Se_6 (**Fig. R6**). The experimental results show that the SHG conversion efficiency increases with the thickness from 12 to 21 nm, while it drops slightly for the 36 nm-thick sample. The result is consistent with the theoretical analysis trend shown in **Fig. R5**. The maximum SHG conversion efficiency of $43.2\%W^{-1}$ can be obtained for a sample with thickness about 21 nm. To maintain consistency, we also replace the values of conversion efficiency and photocurrent with the optimized data in maintext.

Fig. R6 a-c, OM image of SnP₂Se₆ of different thicknesses transferred onto micro ring. **d**, SHG efficiency measured at different thicknesses of SnP₂Se₆

In summary, our device has preliminarily verified the feasibility of integration of frequency doubling conversion and photoelectric detection, but the specific performance indexes need to be further improved. Waveguide structure, Q value of resonant ring, sample thickness and other factors will have a great impact on the performance of the device, which will be the focus of our research in the future.

Fig. R4 has been updated as part of **Fig. 2**, and related discussion is shown as below:

“In contrast, SnP₂Se₆ nanosheet with few layers (< 7L) demonstrates quadratically increased SH response, and the collected SHG peak intensity for multilayer SnP₂Se₆ nanosheet exceeds that of 2H-MoTe₂ with layer number increasing”

“It should be noted that the influence of light re-absorption is being strengthened gradually in the range from 8 L to bulk, thus leading to the deviation from quadratic relationship even with a downward trend.”

Fig. R5-R6 has been added in Supporting Information as **Fig. S21-23**, and related discussion is shown as below:

“We then estimate and optimize the SHG conversion efficiency of SnP₂Se₆ nanosheet integrated on SiN microring resonator (Supplementary Figure S20-S22)”

“The maximum value of 43.2%W⁻¹ can be obtained for a sample with thickness about 21 nm, which is reasonably satisfying for our experimental setup and outstanding in

previously reported TMDCs-integrated photonic structures.”

“The SHG efficiency for a lossless waveguide without pump depletion is given by the following expression:

$$\eta_{TE_0-TM_2} = \frac{P_{SH}}{P_{FF}^2} \times 100\% = \xi_{NL}^2 L^2 \frac{\sin^2(\Delta\beta L/2)}{(\Delta\beta L/2)^2} \times 100\% \quad (S6)$$

The Q value in the micro ring determines the overall length of propagation L , $\Delta\beta = 2\beta^{pump} - \beta^{SHG}$ represents the wave vector mismatch, and ξ_{NL} represents nonlinear overlap factor, which can be defined as:

$$\xi_{NL} = n^2 \chi_{SnP_2Se_6}^{(2)} \sin 3\theta \left(\frac{8\pi^2}{\epsilon_0 c \lambda_{FF}^2 n_{SH} n_{FF}^2} \right)^{\frac{1}{2}} \left(\frac{\int_{SnP_2Se_6} (E_{FFx}^2)^2 E_{SHy} dx dz}{\left(\int_{all} |E_{FF}| dx dz \right)^2 \left(\int_{all} |E_{SH}| dx dz \right)} \right) \quad (S7)$$

In this formular, θ is the angle formed by the guided-mode wave vector and the armchair direction of the SnP_2Se_6 crystal. However, since our device is fabricated based on a micro ring, this term can be eliminated by integral. Considering this, the equation (S7) can be reduced to:

$$\xi_{NL} = n^2 \chi_{SnP_2Se_6}^{(2)} \left(\frac{8\pi^2}{\epsilon_0 c \lambda_{FF}^2 n_{SH} n_{FF}^2} \right)^{\frac{1}{2}} \left(\frac{\int_{SnP_2Se_6} (E_{FFx}^2)^2 E_{SHy} dx dz}{\left(\int_{all} |E_{FF}| dx dz \right)^2 \left(\int_{all} |E_{SH}| dx dz \right)} \right) \quad (S8)$$

where $\chi_{SnP_2Se_6}^{(2)}$ is the second-order nonlinear susceptibility of SnP_2Se_6 , n is the layer of the material, $\lambda_{FF}=1550$ nm is the pump wavelength, n_{FF} and n_{SH} represents the effective refractive indices of the TE_0 mode at fundamental frequency and TM_2 mode at Second-harmonic frequency modes, respectively. ϵ_0 and c are the permittivity and light speed in vacuum. $\int_{SnP_2Se_6}$ and \int_{all} denote two-dimensional integration over SnP_2Se_6 and all space, respectively. E_{FFx} is the x component of $E_{FF}(x, z)$, the electric field of the fundamental mode TE_0 , and E_{SHy} is the y component of $E_{SH}(x, z)$, the

electric field of the second-harmonic mode TM_2 . These electric field distribution data are obtained by Lumerical MODE solutions.

In fact, $n^2\chi_{SnP_2Se_6}^{(2)}$ reflects the SHG conversion intensity of the SnP_2Se_6 with different layers. From **Fig. 2e**, we noted that the SHG intensity reveals a near-quadratic dependence when the layer is less than 7, while the light re-absorption would lead to the deviation from the quadratic relationship, even showing a downward trend in the range from 8 to 70 layers. Therefore, we can obtain the fact value of $n^2\chi_{SnP_2Se_6}^{(2)}$ with dependence on the layer numbers (**Fig. S20**).

By substituting the data from **Fig. S20** into equation (S8) and equation (S6), we can theoretically calculate the SHG conversion efficiency. **Fig. S21** shows the theoretical variation trend of the SHG conversion efficiency with material thickness. Different material thicknesses would result in various propagation constants, electric field distribution of incident waves, and second harmonics for the situation of fixed propagation length and waveguide width, which is the primary cause for the difference in SHG conversion efficiency. Here, the waveguide width was set to be $1.26\ \mu\text{m}$ and the bending radius of the micro ring to $25\ \mu\text{m}$. According to **Fig. S21**, we can conclude that the number of material layers providing the optimal SHG conversion efficiency is around 64 in this device configuration. It's important to note that this value is highly relative to the fixed device. Once the size parameter of the micro ring changes, the optimal conversion layer of the material will also change.

To further prove the above conclusion, we have supplemented the experimental data of SHG efficiency in micro ring structures covered with different thicknesses of

SnP₂Se₆ (Fig. S22). The experimental results show that the SHG conversion efficiency would increase with the thickness of samples from 12 nm to 36 nm, which is consistent with the theoretical analysis trend shown in Fig. S21.

In summary, our device has preliminarily verified the feasibility of integration of frequency doubling conversion and photoelectric detection, but the specific performance indexes need to be further improved. Waveguide structure, Q value of resonant ring, sample thickness and other factors will have a great impact on the performance of the device, which will be the focus of our research in the future.”

References

- 1 Zhang, Y. *et al.* Inversion symmetry broken 2D SnP₂S₆ with strong nonlinear optical response. *Nano Research*, 1-7 (2022)
- 2 Zhao, M. *et al.* Atomically phase-matched second-harmonic generation in a 2D crystal. *Light: Sci. Appl.* **5**, e16131-e16131 (2016).
- 3 Chen, H. *et al.* Enhanced second-harmonic generation from two-dimensional MoSe₂ on a silicon waveguide. *Light: Sci. Appl.* **6**, e17060-e17060 (2017).
- 4 Luo, R. *et al.* Highly tunable efficient second-harmonic generation in a lithium niobate nanophotonic waveguide. *Optica* **5**, 1006-1011 (2018).

REVIEWERS' COMMENTS

Reviewer #1 (Remarks to the Author):

The authors have now addressed all my comments and I can support publication of the manuscript in its current form

Reviewer #2 (Remarks to the Author):

The authors have fully addressed my questions. I recommend it for publication.